# The Role of the Vaginal and Endometrial Microbiomes in Infertility and Their Impact on Pregnancy Outcomes in Light of Recent Literature

**DOI:** 10.3390/ijms252313227

**Published:** 2024-12-09

**Authors:** Bernadett Balla, Anett Illés, Bálint Tobiás, Henriett Pikó, Artúr Beke, Miklós Sipos, Péter Lakatos, János P. Kósa

**Affiliations:** 1Department of Internal Medicine and Oncology, Semmelweis University, 1083 Budapest, Hungary; balla.bernadett@semmelweis.hu (B.B.); barkaszine.dr.illes.anett@semmelweis.hu (A.I.); tobias.balint@semmelweis.hu (B.T.); piko.henriett@semmelweis.hu (H.P.); lakatos.peter@semmelweis.hu (P.L.); 2Hungarian Research Network SE-ENDOMOLPAT Research Group, 1085 Budapest, Hungary; 3Department of Obstetrics and Gynecology, Semmelweis University, 1088 Budapest, Hungary; beke.artur@semmelweis.hu; 4Department of Obstetrics and Gynecology, Assisted Reproduction Centre, Semmelweis University, 1082 Budapest, Hungary; sipos.miklos@semmelweis.hu

**Keywords:** microbiome, bacteria, female genital tract, vagina, endometrium, fertility, pregnancy

## Abstract

The Human Microbiome Project (HMP), initiated in 2007, aimed to gather comprehensive knowledge to create a genetic and metabolic map of human-associated microorganisms and their contribution to physiological states and predisposition to certain diseases. Research has revealed that the human microbiome is highly diverse and exhibits significant interpersonal variability; consequently, its exact impact on health remains unclear. With the development of next-generation sequencing (NGS) technologies, the broad spectrum of microbial communities has been better characterized. The lower female genital tract, particularly the vagina, is colonized by various bacterial species, with *Lactobacillus* spp. predominating. The upper female genital tract, especially the uterus, was long considered sterile. However, recent studies have identified a distinct endometrial microbiome. A *Lactobacillus*-dominated microbiome of the female genital tract is associated with favorable reproductive outcomes, including higher success rates in natural conception and assisted reproductive technologies (ART). Conversely, microbial imbalances, or dysbiosis, marked by reduced *Lactobacilli* as well as an increased diversity and abundance of pathogenic species (e.g., *Gardnerella vaginalis* or *Prevotella* spp.), are linked to infertility, implantation failure, and pregnancy complications such as miscarriage and preterm birth. Dysbiosis can impair the vaginal or endometrial mucosal barrier and also trigger pro-inflammatory responses, disrupting essential reproductive processes like implantation. Despite growing evidence supporting the associations between the microbiome of the female genital tract and certain gynecological and obstetric conditions, clear microbial biomarkers have yet to be identified, and there is no consensus on the precise composition of a normal or healthy microbiome. The lack of standardized protocols and biomarkers limits the routine use of microbiome screening tests. Therefore, larger patient cohorts are needed to facilitate comparative studies and improve our understanding of the physiological microbiome profiles of the uterus and vagina, as well as how dysbiosis may influence clinical outcomes. Further research is required to refine diagnostic tools and develop personalized therapeutic strategies to improve fertility and pregnancy outcomes.

## 1. Introduction

Infertility is defined as the inability to conceive after 12 months of regular, unprotected sexual intercourse, affecting approximately 8.8% of women of reproductive age. Globally, between 50 and 80 million individuals experience infertility, according to World Health Organization surveys [1]. Based on epidemiological data, 85% of infertility cases have an identifiable cause, with ovulatory dysfunction, male factor infertility, and tubal disease being the most prevalent contributors. Among ovulatory disorders—including anovulation—70% are associated with polycystic ovary syndrome (PCOS), accounting for about 25% of all infertility diagnoses. In contrast, 15% of cases are classified as unexplained infertility, where no specific etiology is determined despite thorough evaluation. Lifestyle and environmental factors, primarily smoking, obesity, and advanced age, can further impair fertility outcomes. Around one in eight women aged 15 to 49 seek medical assistance for infertility. While the success rates of treatments depend on age and underlying diagnoses, a combination of accurate diagnostic methods, effective therapies, and collaborative decision-making has proven to help many couples achieve their fertility goals [1,2,3].

To enhance our understanding of the factors influencing infertility, it is crucial to examine the key elements that contribute to fertility. Emerging research increasingly points to a significant correlation between infertility and the microbiota. Studies have shown that infertile women exhibit distinct microbial compositions in the lower and/or upper reproductive tract compared to fertile women. Given these findings, the role of the microbiome must be recognized as a critical factor in reproductive health [4,5].

Our literature review aims to provide a detailed overview of the existing evidence in relation to the female genital microbiome and infertility, as well as its potential implications for assisted reproductive technologies (ARTs).

## 2. The Vaginal Microbiome and Its Impact on Fertility

The healthy vaginal microbiome is well known for being dominated by *Lactobacillus* species, forming what is referred to as Döderlein flora. *Lactobacillus* spp. dominate the lower genital tract at a concentration of 10^7^–10^8^ colony-forming units per gram of vaginal fluid [6]. Different species of *Lactobacillus* produce both hydrogen peroxide and lactic acid. When *Lactobacillus* dominance is maintained, the vaginal pH stays below 4.5, thus maintaining an acidic environment. The homogenous composition and balance of the microbiome in the genitourinary tract, known as eubiosis, is critical for the health of the female reproductive system [7]. The vaginal microbiome dynamically shifts in response to physiological influences, such as the menstrual cycle, pregnancy, and hormonal changes during menopause. In addition, external factors, such as diet, sexual activity, and hygiene practices, also play a role [8]. Estrogen can stimulate the proliferation of the vaginal epithelium and increase glycogen deposits, which favor the growth of glucose-fermenting *Lactobacilli*. Therefore, the normal changes and fluctuations in estrogen levels over the lifespan from puberty to post-menopause or during the menstrual cycle can significantly influence the bacterial content of the vaginal microbiota [9]. Among all medications, antibiotics are the most common cause of disruption to the vaginal flora, leading to a dysbiotic state. Additionally, the use of contraceptives may also disturb vaginal eubiosis. Recent studies have led to significant advances in our understanding of the composition of and the factors influencing the stability of vaginal microbial communities [10].

### 2.1. Community State Types (CSTs) of the Vaginal Microbiome

One of the first comprehensive studies on this topic, published in 2011, involved the phylogenetic analysis of the vaginal microbiome of nearly 400 asymptomatic, reproductive-aged North American women. The women were categorized into four ethnic groups: White (Caucasian), Black (African American), Hispanic (Latina), and Asian. The study found that the vaginal microbial communities could be classified into five distinct groups (I through V) based on their composition and dominant species. Overall, *Lactobacillus* was the dominant genus in 73% of the vaginal microbiome communities. The following four *Lactobacillus* species—*L. crispatus* (group I), *L. gasseri* (group II), *L. iners* (group III), and *L. jensenii* (group V)—were the most prevalent bacteria among asymptomatic, reproductive-aged women. Group I was found in 26.2% of the women sampled [11]. *L. crispatus* is thought to provide the most protective benefits via various mechanisms (e.g., proteinaceous outer surface layer or D-lactic acid production), partly due to its ability to produce hydrogen peroxide, which is toxic for catalase-negative bacteria. While *L. iners* is perhaps the most common vaginal bacteria, longitudinal studies have found that *L. iners*-dominated communities are less stable than those dominated by other *Lactobacilli* [12]. The core genome of *L. crispatus* is nearly twice the size of the *L. iners* genome, and therefore has a wider range of metabolic capabilities. *L. iners* is able to ferment fewer carbon sources and lacks the enzymatic set required for the synthesis of several essential amino acids, thus relying more on exogenous resources. Overall, the limited genomic repertoire of *L. iners* may make this strain more sensitive to environmental fluctuations [13]. It is also confirmed that *L. crispatus* produces a serine protease that can degrade the pro-inflammatory interferon-gamma-inducible protein 10 (IP-10). This mechanism may be involved in the colonization of vaginal epithelial cells by *L. crispatus*; previous in vitro studies have shown that this results in more attenuated inflammatory signaling than colonization with *L. iners* [13]. In addition, *L. iners* produces a pore-forming lysine (inerolysin) known to disrupt the integrity of the host cell membrane [14]. In line with the above, Leizer J and coworkers confirmed that vaginal epithelial cells exhibit higher levels of autophagy and a lower induction of stress-related hsp70 when *L crispatus* is the abundant strain, compared to when *L. iners* dominates [15].

The original study by Ravel J et al. found that in 27% of women, microbial diversity and heterogeneity were characteristic of their communities. In group IV, *Lactobacillus* spp. were underrepresented or entirely absent, and these communities were instead composed of lactic acid-producing obligate and facultative anaerobes, including *Prevotella*, *Dialister*, *Atopobium*, *Gardnerella*, and *Megasphaera* species. Both vaginal pH levels and bacterial communities varied across different ethnic groups. Vaginal pH was higher in Hispanic (pH 5.0 ± 0.59) and African American (pH 4.7 ± 1.04) women compared to Asian (pH 4.4 ± 0.59) and White (pH 4.2 ± 0.3) women. Moreover, group IV was found in more in Latina (34.3%) and African American (38.9%) women than Asian (17.6%) and Caucasian (9.3%) women [11]. An additional observation also suggested that the vaginal microbial ecosystem tends to be more stable when *L. crispatus* is prevalent, as opposed to when *L. iners* is the principal species [13]. It is now known that *L. iners*-dominated group III communities often transition to group IV, which may contribute to their limited association with vaginal health. Previous meta-analyses have indicated that *L. iners* may become a dominant species when the microbiome is in a transitional stage from abnormal to normal. This could help the microbiome restore to a *Lactobacilli*-dominated community after bacterial vaginosis (BV) [16,17].

Based on a comprehensive set of 13,160 taxonomic profiles from a total of 1975 women of reproductive age, slightly different vaginal community state types (CSTs) than previously described were identified by the VALENCIA study led by France MT et al. In this classification system, there are seven CSTs, four of which have a high relative abundance of *Lactobacillus* species. CST I—characterized by a predominance of *L. crispatus*; CST II—dominated by *L. gasseri*; CST III—dominated by *L. iners*; and CST V—characterized by *L. jensenii* dominance. Three additional CSTs termed CST IV-A, IV-B, and IV-C were also defined that did not have a high relative abundance of *Lactobacilli*. CST IV-A contains a high proportion of *Candidatus Lachnocurva vaginae* (formerly known as *BVAB1*), with *Gardnerella vaginalis* and *Atopobium vaginae.* A large percentage of *Gardnerella vaginalis*, with *Candidatus Lachnocurva vaginae, Atopobium vaginae,* and *Prevotella* spp., belong to CST IV-B. CST IV-C is composed of *Lactobacillus* spp., *Gardnerella vagnalis*, *Atopobium vaginae,* and *Candidatus Lachnocurva vaginae*. A further five subtypes can be distinguished according to the dominant species: C0—balanced community with minimal *Prevotella* spp. presence; C1—*Streptococcus* spp. dominance; C2—*Enterococcus* spp. abundance; C3—*Bifidobacterium* spp. dominance; and C4—*Staphylococcus* spp. dominance. Black women were more likely to have CST IV-A than White women (*z* = 4.9, *p* < 0.001). CST IV-B was more common in Hispanic women than White women (*z* = 3.3, *p* = 0.006). Finally, the study found that Asian women were more likely to have CST III than both African American and White women (*p* = 0.025; *p* = 0.050) [18]. The vaginal microbiota has been observed to undergo fluctuations in some women, including shifts in community state types (CSTs), particularly during the onset of menstruation or following unprotected vaginal intercourse. Other alterations in the vaginal microbiota cannot be attributed to a specific cause with certainty, and may result from variations in host physiology or competitive interactions within the microbial community. However, in some individuals, the vaginal microbiota has been shown to remain stable over time, maintaining a consistent community composition across multiple menstrual cycles [19]. Based on the results, it appears that CSTs I, III, and IV are the most prevalent and account for approximately 90% of reproductive-age women. Samples of different vaginal community state types (CSTs) from asymptomatic women of reproductive age are shown in Figure 1. Although the CST approach does simplify community composition, it continues to be an important framework for the study of the vaginal microbiota.

There is currently no consensus on how individual CSTs affect fertility and pregnancy rate. The studies carried out have produced divergent results. In the work of Lledo B et al., no significant differences in the distribution of CSTs were reported in women, regardless of spontaneous or IVF pregnancies [20]. The study of Koedooder R et al. revealed that women with CST I or CST III have higher chances of becoming pregnant following embryo transfer compared to women with vaginal microbiotas belonging to CST IV or CST V [21]. Fertile women mostly present with *Lactobacillus*-dominated CSTs, with only low numbers of diverse communities presenting the *Gardnerella*-dominated CST IV. The proportion of diverse communities is enhanced in both infectious and non-infectious infertilities [22]. In line with this, Haahr T. et al. showed that CST IV was associated with a higher Nugent score, indicating increased likelihood of bacterial vaginosis compared to all other CSTs. Women with CST IV were less likely to experience clinical pregnancy [23].

### 2.2. Vaginal Microbiome and Fertility Outcomes

The protective role of the vaginal microbiome has become increasingly evident, and numerous studies have shifted focus to understanding the connections between bacterial vaginosis (BV), urinary tract infections, pelvic inflammatory disease (PID), sexually transmitted infections (STIs), and dysbiotic vaginal flora. Dysbiosis can occur due to changes in overall microbial diversity, i.e., a loss of beneficial microorganisms or an overgrowth of potentially harmful organisms [24]. According to international consensus, bacterial vaginosis is a pathological dysbiotic condition characterized by the depletion of the normal *Lactobacillus* flora. During this state, lactic acid bacteria are replaced by a mixed community predominantly consisting of *Gardnerella vaginalis*, *Atopobium vaginae*, *Mycoplasma hominis*, *Peptostreptococcus*, and *Prevotella* species [25]. A wide variety of bacteria can cause dysbiosis, though none play an exclusive role, and most can also be detected in small numbers in the vaginal secretions of healthy women. BV occurs in 20–30% of reproductive-aged women, and the complications of both BV and vaginal dysbiosis include preterm birth, increased susceptibility to STIs, chronic pelvic pain, and in severe cases, infertility [26]. BV without STI coinfection was found in approximately 19% of infertile individuals [6]. Imbalance of the lower genital tract microbiota has been coupled with an elevated risk of pelvic infections. The presence of *E. coli* and other BV-associated pathogens, such as *Atopobium vaginae*, *Sneathia sanguinegens*, *Sneathia amnionii*, *Chlamydia trachomatis*, *Mycoplasma genitalium,* and *Neisseria gonorrhoeae*, can result in pelvic inflammatory disease (PID), affecting the fallopian tubes, ovaries, uterus, and pelvic peritoneum. Due to the resulting chronic inflammation and fibrinous adhesion inside the fallopian tubes, conception is impaired; this is also one of the main causes of tubal infertility and ectopic pregnancy [27].

Dysbiosis of the vaginal microbiota encompasses a whole spectrum of changes in local immune processes. It is characterized not merely by a reduction in lactic acid-producing *Lactobacillus* spp. and an excessive growth of various anaerobes, but also by an increase in mucin-degrading enzymes (e.g., sialidase and prolidase) and cytotoxins (e.g., vaginolysin). In addition, a pro-inflammatory response is induced with the increased secretion of interleukin-1a (IL-1a), interleukin-1b (IL-1b), interferon, and tumor necrosis factor; neutrophil infiltration occurs and the mucosal barrier becomes impaired, as well. The inflammatory milieu resulting from dysbiosis may be responsible for the development of different infertility problems; among others, altering the viscosity of the cervix, thereby impeding sperm passage, and also impairing sperm motility and viability. It can also adversely affect the quality and quantity of eggs released during ovulation and the process of implantation, and finally, it makes the cervix weaker and more prone to premature opening, thus increasing the risk of miscarriage [28,29,30].

A Danish research group was among the first to investigate the potential role of the vaginal microbiome in infertility. In their 2016 published findings, they demonstrated that bacterial imbalances in the vaginal microbiome influence pregnancy rates and the success of infertility treatments [31]. According to their data, vaginal dysbiosis (VD) was found in 28% of 130 women undergoing in vitro fertilization (IVF). In addition, bacterial vaginosis was confirmed in 21% of these cases. These conditions often present without symptoms, making detection difficult. In women with an abnormal vaginal microbiome (dysbiosis or BV) identified during assisted reproductive procedures, only 9% achieved clinical pregnancy. In contrast, women with a normal vaginal microbiome had a significantly higher pregnancy success rate of 44%. In summary, BV is associated with a lower implantation rate and an increased risk of early pregnancy loss. Celicanin MM et al. demonstrated in their meta-analysis that early pregnancy loss had a significantly higher prevalence in patients with vaginal dysbiosis (bacterial vaginosis and aerobic vaginitis) compared to patients without VD. Overall, patients with VD undergoing IVF have a markedly lower clinical pregnancy rate per embryo transfer than women without VD [32]. Supporting this, a cross-sectional study by Babu G et al. (2017), involving 84 healthy women and 116 women with reproductive failure, suggested that routine vaginal microbiome screening should be introduced for all women undergoing infertility treatments [33]. In the 2024 study by Chopra C et al., notable reduction in the prevalence of *Lactobacillus* was observed in women suffering from idiopathic infertility. The analysis of the vaginal microbiome revealed a significantly increased presence of *Gardnerella*, *Prevotella*, *Atopobium,* and *Enterococcus*, whereas a parallel decrease in the relative abundance of *Firmicutes* was detected in infertile patients [34]. It is well established that contraceptive use can alter the vaginal bacterial flora. For example, women using a pessary show increased populations of *Escherichia coli*, *Enterococcus*, and other Gram-negative bacteria, while oral contraceptives have been shown to decrease *E. coli* and *Candida* spp. in the vaginal microbiota [35].

Furthermore, it has been demonstrated that the absence of *Lactobacilli* dominance in the vagina during pregnancy is associated with an increased risk of preterm birth. Bacterial vaginosis, aerobic vaginitis, transitional flora, and a low count of *Lactobacillus* spp. are all predisposing factors for preterm delivery [36]. It is also well known that pregnant women with BV are at increased risk of chorioamnionitis, miscarriage, preterm birth, and premature or early rupture of the amniotic sac [37]. Kuon RJ et al. suggest a link between vaginal infections and the tendency of recurrent pregnancy loss (RPL). Their research on 243 patients with a history of three or more consecutive miscarriages revealed the presence of *Gardnerella vaginalis* (19%) and Gram-negative anaerobic bacteria (20.5%); furthermore, in 14.5% of the cases, commensal *Lactobacilli* were not present [38]. This was strengthened by the work of Su W et al., who found that vaginal samples exhibited a higher volume of *Lactobacillus* and a lesser abundance of *Gardnerella* in the group of healthy controls and women showing implantation success, in comparison to women with implantation failures [39]. Despite the ever-growing number of studies using modern molecular genetic technologies to investigate the vaginal microbiome during pregnancy, no definitive microbial marker has yet been identified that is clearly associated with preterm birth [40].

A recent study highlighted the association between vaginal microbiome imbalance and the development of malignant tumors. Persistent human papillomavirus (HPV) infection is a well-established causal factor in the development of cervical cancer. However, much remains unknown about the composition and function of the microbiome in the context of HPV infection. It was found that in the HPV16-positive group, the relative abundance of dominant *Lactobacillus* and *Firmicutes* species was lower. Researchers concluded that vaginal dysbiosis may be associated with HPV infection in the female genital tract and, in the long term, the development of cervical cancer [41]. It is also known that BV can increase the risk of HPV infection or reactivation [42].

Infertility, a significant public health concern, has a global prevalence of 17.5% among couples of reproductive age. Approximately 30% of cases, according to the literature, are still classified as unexplained or as idiopathic infertility, i.e., when no causative factors can be identified despite comprehensive medical evaluation [34]. Given the effect of the vaginal microbiome on reproductive health, a thorough understanding of the complex aspects of this relationship could offer future solutions for explaining and treating idiopathic cases.

## 3. Vaginal Microbiome and Pregnancy

The composition of the vaginal microbiota is strongly influenced by pregnancy and may contribute to adverse pregnancy outcomes, such as spontaneous preterm birth. The vaginal microbiota in pregnant women is less rich and less diverse compared to non-pregnant ones, with a predominance of Lactobacillus species and fluctuations in species stability influenced by gestational age [43]. An *L. crispatus*-dominated vaginal microbiome is the most common type during pregnancy, particularly in healthy pregnant women [44,45,46]. In a study examining a diverse cohort of women during their final trimester of pregnancy, *Lactobacilli*, including *L. crispatus*, were shown to play a critical role in maintaining low microbial diversity [47]. Key factors affecting the vaginal microbiota during pregnancy include individual genetic makeup and gestational age. Additional factors influencing the vaginal microbiota include maternal age, parity (number of previous pregnancies), obesity, and self-reported cannabis use [48]. In uncomplicated pregnancies, the vaginal microbiota remains in or transitions to a *Lactobacillus*-dominated community state [49]. Other bacterial species commonly present in the vaginal microbiome include *Gardnerella vaginalis, Atopobium vaginae, Prevotella* spp., *Sneathia amnii,* and *Candidatus Lachnocurva vaginae* [50]. Several studies have consistently shown that increased microbial diversity has been linked to pregnancy complications. Walther-Antonio MR et al. [51], who collected vaginal samples every eight weeks, characterized the normal pregnant vaginal environment by low diversity and high stability. Similarly, DiGiulio DB et al. observed that the diversity and composition of the vaginal microbiome remained stable throughout pregnancy, even in cases of preterm birth [52].

Changes in vaginal microbial composition may also be associated with conditions such as preterm birth, miscarriage, preeclampsia, ectopic pregnancy, gestational diabetes mellitus, and chorioamnionitis [53]. In a prospective study, an increased prevalence of CST IV was observed among preterm birth (PTB) cases [54]. However, contradicting results are found in the recent literature. According to certain studies, most normal term cases are associated with CST IV, CST I, or CST II, while PTB cases are assigned to CST IV, CST III, or CST V [55,56,57,58]. In contrast, no correlation was found between specific CSTs and PTB in some publications [59,60,61]. Since increased alpha diversity is often the initial indicator of significant microbial differences, many studies use it as an initial analytical step. Generally, the vaginal microbiome in healthy, uncomplicated pregnancy cohorts shows low diversity and *Lactobacillus* dominance. The overall diversity in microbial composition is higher in PTB and serves as a prognostic factor. In addition, the relative abundances of several taxa, including *BVAB1*, *Gardnerella* spp., *Atopobium* spp., *Prevotella* spp., *Sneathia* spp., and *Megasphaera* spp., have been linked to this pathological condition [45,54,62,63,64,65,66]. Women delivering at term were found to be less likely to be positive for *Mycoplasma* spp. and *Ureaplasma* spp. [67].

Several studies have aimed to elucidate the relationship between vaginal microbial composition and miscarriage or recurrent pregnancy loss (RPL). An analysis of alpha and beta diversity in miscarriage cases also revealed increased microbial diversity [53,68]. A large prospective study found that miscarriage was linked to *Lactobacillus* depletion, with a higher prevalence of CST IV [69]. In a prospective study, women who experienced miscarriage showed a greater presence of *Mycoplasma genitalium* and *Ureaplasma,* especially *U. parvum* [58,70]. In addition, specific associations have been found between the presence of *Atopobium* and RPL, with a relative abundance of over 0.01% of *Atopobium* proposed as a potential biomarker for predicting first-trimester spontaneous miscarriage [71,72].

While the role of a balanced vaginal microbiome in preventing gynecological disorders and maintaining reproductive health is not yet fully understood, an abnormal vaginal microbial composition appears to increase the risk of miscarriage, PTB, RPL, and recurrent implantation failure (RIF) [73]. *Lactobacillus* species are the most common bacteria in healthy female reproductive tracts. However, the presence of *L. iners* in women with RIF and unexplained infertility suggests this species may play an unfavorable role in fertility outcomes [74]. Future research should focus on species-level analysis to better understand the specific roles of different *Lactobacillus* strains in reproductive outcomes [75]. Microbiome tests conducted prior to conception and during early pregnancy could help identify microbial compositions conducive to successful embryo implantation and placenta formation, and may potentially reduce miscarriage rates [68].

## 4. The Endometrial Microbiome and Its Role in Fertility

There is growing evidence that the uterine microbiome plays a crucial role in women’s reproductive health, particularly in maintaining pregnancy. Dysbiosis in uterine flora may reduce the success rates of assisted reproductive technologies, among other complications [76,77].

### 4.1. Bacterial Composition of the Endometrial Microbiome

To date, several studies have characterized the microbiota profile of the endometrium in reproductive-aged women considered normal or healthy, with most reporting a dominance of *Lactobacillus* species. However, it is important to note that there is still no consensus on a core microbiota. In a study examining the upper genital tract, based on an analysis of 58 endometrial samples, *L. iners*, *Prevotella* spp., and *L. crispatus* were found in high abundance in 95% of the patients [78]. Later research using endometrial fluid samples from 13 reproductive-aged women similarly demonstrated the abundant presence of *Lactobacillus* species. These findings suggest two potential types of bacterial composition in the endometrium: *Lactobacillus*-dominant (*Lactobacillus* spp. > 90%) and non-*Lactobacillus*-dominant (*Lactobacillus* spp. < 90%). Other genera frequently detected in endometrial fluid samples include *Bifidobacterium*, *Gardnerella*, *Prevotella*, and *Streptococcus* [79]. The non-*Lactobacillus*-dominated (NLD) endometrial microbiota is associated with significant decreases in implantation, pregnancy, and live birth in infertile women undergoing assisted reproduction treatment, whereas a *Lactobacillus*-dominant (LD) microbiota is advantageous for embryo implantation [79]. Hashimoto T et al. defined the eubiotic stage of the endometrial microbiome as *Lactobacillus* + *Bifidobacterium* spp. ≥80% and the dysbiotic stage as *Lactobacillus* + *Bifidobacterium* spp. <80% with ≥20% other bacteria [80]. A recently published study confirmed that *Lactobacillus* is the most prevalent genus in the endometrium, alongside other bacteria such as *Anaerococcus*, *Atopobium*, *Bifidobacterium*, and *Gardnerella*, though these were present in significantly smaller proportions [77].

In general, endometrial communities associated with dysbiosis show higher bacterial diversity than the unique *Lactobacilli*-dominated eubiotic state. Dysbiosis is also characterized by a lower percentage or complete lack of *Lactobacillus* spp., in addition to higher rates of single or coexisting pathogen species (*Gardnerella*, *Prevotella*, *Mycoplasma*, *Atopoboum* spp.) as well as the overgrowth of commensal bacterial genera, as shown in Figure 2.

### 4.2. Endometrial Microbiome and Fertility Outcomes

Similar to the vaginal microbiome, endometrial bacterial communities are also influenced by factors such as age, hormonal changes, ethnicity, sexual activity, and intrauterine device use [81]. Observations show that microbial diversity within individual samples (alpha diversity, meaning the variety of bacteria constituting the endometrial community) decreases with age, while similarity between individuals (beta diversity) is higher among women 20 years of age or younger [76]. A higher number of abortions and vaginal births can reduce the differences between the endometrial and vaginal microbiomes by disrupting the closed uterine environment through cervical incompetence. Under certain conditions, changes in the vaginal microbiome can directly affect the bacterial communities of the endometrium [82]. This theory is supported by findings that women with a non-*Lactobacillus*-dominant (NLD) endometrial microbiome often also have an NLD or dysbiotic vaginal microbiome as well [83]. The endometrial microbiome is also significantly influenced by hormonal changes [84]. For example, exogenous progestin administration reduces the diversity of *Lactobacillus* spp. phylotypes. Following controlled ovarian stimulation and exogenous progesterone administration, the abundance of *Atopobium* and *Prevotella* species has been observed to increase. Natural hormonal fluctuations during the menstrual cycle also cause characteristic changes. During the proliferative phase, bacterial growth intensifies, leading to increased peptidoglycan and aminoacyl-tRNA synthesis. Additionally, the proportion of *Lactobacillus* is low after menstruation but gradually increases during follicular development, peaking in the luteal phase [85].

Alterations in the uterine microbiome can contribute to reproductive difficulties and infertility through multiple pathomechanisms. The presence of pathogenic or non-commensal bacterial populations may compromise the integrity of the endometrial mucosal barrier by disrupting epithelial tight junctions and reducing the secretion of antimicrobial peptides and mucins. This disruption weakens the host’s defense mechanisms, and the invasion of pathogen species elicits a strong immune response, leading to the overexpression of pro-inflammatory cytokines in the endometrial environment [86]. This reactive environment, burdened with inflammatory processes, has a known negative impact on reproductive outcomes. Some hypotheses suggest that a decrease in *Lactobacillus* abundance results in a significant elevation in the pH of the female genital tract, impairing the ability of embryos to implant. Inflammatory responses triggered by changes in the bacterial milieu have also been associated with fertility problems. Elevated concentrations of total IgM, IgA, and IgG have been observed in chronic endometritis (CE) and recurrent implantation failure (RIF), as well as significantly elevated cytokine levels, including interleukin-6 (IL-6) and pro-IL-1β, which have also been observed in these clinical conditions [87]. Separate studies have revealed that CE is allied with a significant reduction in alpha diversity within the endometrial microbiota and the overexpression of IFN-γ and tumor necrosis factor-α in apoptotic pathways. The implantation endometrium exhibits robust antimicrobial effects; regulatory T cells responsible for immune tolerance and chemokine ligand 14 secretion help maintain epithelial integrity and protect the developing conceptus. These factors and physiological immunological processes can be suppressed in a dysbiotic endometrial environment [88]. Given the limited knowledge about the interaction between the endometrial microbiome and the host’s immune response, further studies are needed to clarify how specific bacterial species might modify inflammatory processes. The role of *Lactobacillus* spp. in the regulation of inflammation in the vaginal microenvironment has previously been described, and it is assumed that *Lactobacillus* may contribute to uterine homeostasis by inducing the secretion of certain anti-inflammatory cytokines, such as *interleukin-1 receptor antagonist* (IL-1RA) [89]. In another study, the concentration of pro-inflammatory factors, i.e., IL-1β, IL-6, hypoxia-inducible factor 1-alpha (HIF-1α), and cyclooxygenase-2 (COX-2), was inversely related to uterine *Lactobacillus* abundance [90]. Endometrial dysbiosis is associated with recurrent implantation failure (RIF). Microbial diversity in the endometrium is significantly higher in women experiencing RIF. Specific bacterial groups, such as *Burkholderia,* and genera like *Atopobium*, *Delftia*, *Gardnerella*, and *Prevotella* that were absent in healthy controls showed significantly higher abundance in women with RIF compared to those unaffected by this medical condition [91].

A prospective study found that a higher abundance of *Lactobacillus* in the endometrium was associated with higher live birth rates. This finding suggests that a healthy, *Lactobacillus*-dominant (LD) endometrial microbiome could serve as a biomarker for IVF success. In contrast, the presence of *Atopobium*, *Bifidobacterium*, *Chryseobacterium*, *Gardnerella*, *Haemophilus*, *Klebsiella*, *Neisseria*, *Staphylococcus*, and *Streptococcus* was associated with poorer clinical outcomes following assisted reproductive technologies. These adverse reproductive outcomes included failure to achieve pregnancy, biochemical pregnancy, and early spontaneous miscarriage. *Gardnerella*, *Klebsiella*, and *Streptococcus* were found to be significantly elevated in women who did not conceive, while *Klebsiella* and *Staphylococcus* were associated with clinical miscarriage [77].

In a research study published in 2023, the abundance of *Streptococcus*, *Dialister,* and *Prevotella* genera was negatively correlated with *Lactobacillus* spp. in RIF patients. Conversely, *Lactobacillus* spp. was consistently enriched in women reaching live birth [92]. According to culturomics technology, associations have been observed between the presence of *Staphylococcaceae* and *Enterobacteriaceae* families and poor IVF outcomes after blastocyst transfer. Furthermore, *Staphylococcus* spp. was significantly more prevalent in endometrial samples from women experiencing implantation failure. The phylum *Actinobacteria*, specifically the families *Bifidobacteriaceae*, *Corynebacteriaceae*, and *Microbacteriaceae*, was exclusively detected in women who did not achieve clinical pregnancy [93].

## 5. Endometrial Microbiome and Pregnancy

Based on the current literature, a direct link between the composition of the endometrial microbiome, implantation success, and miscarriage may be inferred [88]. While studies indicate that a *Lactobacillus*-dominant (LD) microbiome is strongly associated with reproductive success, the cause-and-effect relationship remains unclear and requires further investigation. The commensal endometrial microbiome may form a symbiotic relationship with the endometrium and its immune mediators, promoting implantation and modulating immune tolerance. Disruptions in this balance could impact endometrial homeostasis, potentially leading to pregnancy loss or implantation defects [88]. A successful implantation is commonly associated with an LD microbiome, whereas a non-*Lactobacillus*-dominant (NLD) microbiome may trigger an inflammatory response that disrupts embryo implantation [94]. Increasing *Lactobacillus* levels to over 90% appears to improve implantation outcomes. *Lactobacillus* species proliferate in specific substrates within the vaginal and endometrial microenvironments, which may also play a crucial role in early embryonic survival during the pre-implantation period [95]. Furthermore, the presence of pathogenic bacteria (*Streptococcus*, *Staphylococcus*, *Neisseria*, and *Klebsiella)* and the depletion of beneficial commensal bacteria could weaken the endometrial mucosal barrier and compromise the integrity of the endometrial epithelium, potentially leading to implantation failure [96,97]. In patients undergoing IVF treatment, regardless of pregnancy outcomes, *Lactobacillus* and *Flavobacterium* were identified as the dominant species [98]. After an IVF process, clinical miscarriage was correlated with a significantly elevated relative abundance of *Haemophilus* and *Staphylococcus*, while higher probability of live birth was linked to a *Lactobacillus*-enriched endometrial microbiome [77]. In another investigation, women who achieved clinical pregnancy presented a higher abundance of *Lactobacillus*, *Anaerobacillus*, *Burkholderia*, and *Gardnerella*, while the non-pregnant group had greater levels of *Delftia*, *Prevotella*, *Ralstonia*, and *Streptococcus* [99]. Interestingly, other studies have reported successful pregnancies with very low *Lactobacillus* concentration, including cases where these bacteria were completely absent [80,100,101]. A large difference was reported in the abundance of *L. iners* between women experiencing miscarriage and those with normal pregnancies [102]. Women experiencing RPL showed a drastically lower relative quantity of *L. crispatus* and higher levels of *L. iners* and *G. vaginalis* compared to healthy controls [103]. *L. iners* has been suggested to be correlated with endometrial dysbiosis and adverse reproductive outcomes, including subfertility and a history of pregnancy loss [104,105]. The microbial profile of endometrial samples collected prior to spontaneous abortion exhibited greater bacterial diversity, including genera such as *Streptococcus*, *Pseudomonas*, and *Propionibacterium*, alongside a reduced abundance of *Lactobacillus* compared to endometrial fluid from healthy pregnancies [106]. The composition of the endometrial microbiome throughout the menstrual cycle differs between women with and without RPL [107]. In one study, it was found that the endometrial microbiome maintained high diversity and richness throughout the menstrual cycle in women with RPL; in contrast, microbial diversity decreased around ovulation and stabilized during the luteal phase in the control group without RPL [107]. This difference in microbiome dynamics might be a distinguishing factor in distinguishing between women with and without RPL.

In the near future, analyzing the upper genital tract microbiota at taxa-level resolution may be critical for identifying true pathogenic bacteria in the endometrium and avoiding unnecessary interventions aimed at NLD microbiomes. It is likely that clinical outcomes in assisted reproductive treatments (ART) are influenced by the combined impact of bacterial communities in the uterine cavity and their interactions with host tissues. This suggests that similar bacterial profiles may lead to different fertility and pregnancy outcomes in different patients. Consequently, a healthy endometrial microbiota might be viewed as a diverse composition of bacterial communities that support embryo implantation and pregnancy maintenance, even with a minor presence of pathogenic bacteria [76]. However, because of the heterogeneity of individual microbiomes and the presently available highly variable research findings, at the moment, it is still uncertain whether optimizing the female genital tract microbiota could enhance ART efficiency, and if so, to what extent [108].

## 6. Interactions Between the Endometrial and Vaginal Microbiomes

Approximately 9% of the human microbiome is localized to the urogenital tract [109]. The lower female genital tract is characterized by a bacterial biomass dominated by *Lactobacillus* species. The uterine cavity harbors a microbial population that is 1000 to 10,000 times smaller than the vaginal microenvironment. Although Lactobacilli are a dominant component, the endometrium exhibits significantly higher diversity than the vagina [110]. The process of uterine colonization is not fully understood. Some hypotheses suggest that microbial transmission could occur via the gut, oral cavity, or bloodstream, or via ascension from the vagina. Colonization may also occur during the use of assisted reproductive technologies or during gynecological procedures, such as intrauterine device (IUD) placement, and microorganisms are also known to attach to sperm [111,112,113]. The ecological stability of both the vaginal and endometrial microbiomes plays a crucial role in the health of the female reproductive system [114]. Bacteria colonizing the vagina can ascend through the cervix to the uterine cavity. However, the microbial interaction between these two anatomical regions remains unclear. Certain studies have suggested that the dysbiotic states of the vaginal and endometrial microbiomes are interconnected and undergo synchronous changes, while other research does not support the notion of parallel, dynamic alterations in these microbial communities [82].

In conclusion, a dysbiotic vaginal microbiome and the associated pro-inflammatory responses may damage the epithelial barrier of the cervix, facilitating bacterial translocation into the endometrium. Vaginal dysbiosis and the resulting inflammatory mechanisms can cause cell-specific changes, such as epithelial-to-mesenchymal transition (EMT) or cell death (apoptosis, necrosis, and autophagy), which contribute to cervical damage and cervical remodeling. Ascending infections from the vagina into the cervicovaginal space significantly increase its IL-6, IL-1β, IL-8, and IL-10 production, and this altered pro-inflammatory cytokine network can disrupt the cervical mucosal barrier [115]. It appears that similar mechanisms contribute to the development of vaginal and/or endometrial dysbiosis, characterized by reduced *Lactobacillus* abundance, the overgrowth of pathogenic species, and increased diversity in the endometrial microbiome—factors that have been linked to implantation failures and miscarriages [75].

Currently, detailed molecular analyses of the female genital tract microbiome, using methods such as next-generation sequencing (NGS) or polymerase chain reaction (PCR), are available for clinical practice. While analysis of the endometrial bacterial flora requires uterine lavage fluid or endometrial tissue biopsy, the vaginal microbiome can be tested using a simple, non-invasive vaginal swab. Extensive literature exists linking the dysbiosis of both the vaginal and endometrial microbiomes—independently—with reproductive failures. Therefore, examining both microbiomes can provide insight into the potential microbial components contributing to female infertility [114].

## 7. Basic Treatment Options for Endometrial and Vaginal Dysbiosis

In a normal, *Lactobacillus*-dominant vaginal microbiome, lactic acid is present at an average concentration of 110 mM. In dysbiosis, however, lactic acid levels often drop below 20 mM, while other organic acids that are weaker than lactic acid (short-chain fatty acids—SCFAs), such as acetic acid, propionic acid, and butyric acid, appear [116]. These alterations in acid profile, along with the presence of various amines, contribute to an increase in vaginal pH. Lactic acid, considered a postbiotic metabolite, has been the focus of numerous studies for the treatment and prevention of bacterial vaginosis. Study results indicate that the intravaginal application of lactic acid—typically once daily for 5 to 7 days—can alleviate symptoms associated with bacterial vaginosis, either on its own or in combination with pharmacological treatments. The use of lactic-acid-based intravaginal products can also promote the recolonization of the vagina by *Lactobacillus* species; in one study, 83% of women treated with lactic acid experienced recolonization compared to just 10% of those receiving a placebo [116]. Vaginal eubiosis can be restored through the use of various *Lactobacillus*-containing probiotic products (e.g., vaginal suppositories, capsules), which have demonstrated efficacy in treating bacterial vaginosis in several clinical trials. *Lactobacillus* spp. supplementation has proven effective in preventing recurrent infections following standard metronidazole therapy [117]. The standard-of-care treatment for bacterial vaginosis includes the use of metronidazole or clindamycin, although this often fails to produce a lasting resolution of the condition [19]. As antibiotics are non-specific and reduce both *Lactobacillus* populations and other commensal bacteria, the simultaneous use of antibiotics and probiotics may represent an optimal therapy for both eradicating pathogens and restoring the vaginal microbiome. Probiotics have also demonstrated potential in preventing preterm birth. Kirihara N and colleagues reported that a combination of oral probiotics (*Streptococcus faecalis*, *Clostridium butyricum*, and *Bacillus mesentericus*) significantly improved perinatal outcomes [118]. In the clinical setting, there is significant interest in improving endometrial dysfunctions to treat uterine dysbiosis and improve infertility treatment outcomes. However, there is no unified protocol for assessing the endometrial microbial composition, nor for the treatment of uterine dysbiosis [119]. Chronic endometritis (CE) is a gynecological condition believed to benefit the most from the modulation of the microbial pattern using antibiotics, as the bacterial infection underlying its pathomechanism is well understood. CE is a persistent inflammatory condition of the endometrial lining, and is often asymptomatic. CE is observed in approximately 10–11% of the general population, with around 25% of cases exhibiting non-specific clinical symptoms. Nowadays, evaluation for CE is still controversial due to the absence of standardized screening methods and an unclear consensus on the benefits of treatment. The lack of universally accepted histological criteria contributes to variability in diagnostic practices and clinical management [120,121]. Probiotic interventions have been extensively investigated in relation to systemic inflammation. Recent studies have investigated *Lactobacillus* strains such as *L. rhamnosus BPL005* for their capacity to lower pH and produce short-chain fatty acids (SCFAs) and lactate, which act against pathogen colonization. Additionally, *L. reuteri RC-14* and *L. rhamnosus GR-1* have been shown to improve the barrier function of endometrial epithelial cells, helping to protect against infection [119]. For a considerable time, the role of CE in infertility was underestimated, but it is now recognized as a potential cause of not only infertility, but also early miscarriages. The observed occurrence of CE in individuals with RIF ranges from 7.7% to 66%, while it is reported to be 2.8% among those with general infertility [122]. CE results in a non-receptive endometrium, which is a major contributor to implantation failure. The efficacy of antibiotic therapy in treating women with chronic endometritis and recurrent implantation failure is well established. For example, amoxicillin therapy has been shown to reduce the risk of bacterial vaginosis and resolve CE symptoms, and has been associated with higher pregnancy and live birth rates [123]. Furthermore, intrauterine infusions of antibiotics after poor oral antibiotic therapy outcomes have also been applied in order to restore the physiological condition of the uterus in CE [124].

Even when antibiotics are used, whether in combination with probiotics or not, they are not always effective. Therefore, other strategies, such as microbiome transplantation, may represent a promising therapeutic approach. While little is known about microbiome transplantation in the female genital tract, similar to fecal transplantation, it may prove beneficial for gynecological and obstetric conditions where patients do not respond to standard antibiotic or probiotic therapies. In the case of endometrial and vaginal microbiome transplantation, future studies will need to determine key parameters, such as dosage, method of administration, optimal microbiota composition, and criteria for selecting appropriate donors. In addition, more research is needed to establish the long-term safety and effectiveness of this medical procedure. The primary risk of female genital tract microbiota transplantation lies in the potential transmission of pathogenic and opportunistic microorganisms. Indeed, safety considerations remain a significant limitation of its in clinical application. The rigorous screening of donors is crucial to minimize the likelihood of transmitting infectious agents. Nevertheless, this technique holds potential as a future treatment to restore microbial eubiosis [125,126].

## 8. Modern Examination Methods for the Vaginal and Endometrial Microbiomes

Traditional, culture-based methods have the advantage of being relatively inexpensive; however, they are difficult to standardize and are highly dependent on the expertise of the individuals conducting the tests and interpreting the results. These methods are also time-consuming and require specific growth media for bacteria to proliferate. Additionally, many bacteria are either non-cultivable or difficult to culture, and some species can rapidly increase in number, leading to unreliable conclusions [127,128]. It is noteworthy that more than 25% of prokaryotic species remain uncultured under routine laboratory conditions [129].

Quantitative polymerase chain reaction (qPCR) technology detects microbial genetic material, allowing the direct presence of microbes to be identified in the sample. It is suitable for both rapid quantitative analysis and typing. Nucleic acid-based tests streamline the diagnostic process by reducing the reliance on labor-intensive clinical criteria. The appearance of culture-independent molecular diagnostics has significantly eased the detection of non-cultivable bacterial species in the field of BV diagnostics [130]. A limitation, however, is that it often cannot distinguish between viable and dead microorganisms. Despite this, qPCR can detect bacterial types that would otherwise not be visible with microscopic or culture-based methods or remain below the detection threshold. The high sensitivity of this method allows for the detection of asymptomatic infections [131].

One of the most advanced available technologies today is next-generation sequencing (NGS). The rapid advancements in NGS and bioinformatics, which have significantly reduced both cost and time requirements, have been major drivers of recent breakthroughs in microbiome research. While 16S rRNA gene sequencing has been the more commonly used approach, shotgun metagenomics is becoming increasingly accessible and popular. The 16S rRNA encoding gene is commonly used for bacterial identification, as well as for studying phylogeny and taxonomy. This gene is present in virtually all bacteria and archaea, and contains both conserved and variable regions, the latter of which can be used to differentiate and classify bacterial strains [132]. Moreover, 16S rRNA sequencing is a type of amplicon sequencing that targets and reads a specific, hypervariable region of the 16S rRNA gene [73]. It is crucial to consider which hypervariable regions are suitable for identifying vaginal and endometrial bacterial species or genera, as there are differences in the resolution of hypervariable regions. The output of 16S rRNA sequencing generates sequencing reads that undergo a series of bioinformatic steps. These pipelines filter out sequencing errors and questionable reads to ensure high-quality data for alignment, which are then used to accurately identify and profile the bacteria present in the samples.

Shotgun metagenomic sequencing involves fragmenting DNA into smaller fragments, which are then sequenced. The resulting fragments are subsequently aligned using bioinformatics tools to identify the species and genes present within the sample [133]. In contrast to 16S rRNA sequencing, which targets a specific DNA region, shotgun metagenomics sequences all genomic DNA in a sample. This allows for the simultaneous identification and profiling of bacteria, fungi, viruses, and other microorganisms. Additionally, because entire genomes are sequenced, it becomes possible to identify and analyze the microbial genes present in the sample (the metagenome), providing insights into the functional potential of the microbiome. Evidence from large-scale human microbiome studies suggests that functional metagenomic data may provide greater power in distinguishing between ‘healthy’ and ‘dysbiotic’ microbiomes [134]. Shotgun sequencing requires more complex bioinformatics methods to process and analyze data [73]. The final output includes details on the relative abundances of various microorganisms and the relative abundances of microbial genes (e.g., those related to metabolism or antibiotic resistance). Table 1 highlights key features to consider when choosing a sequencing method for microbiome studies.

Short-read sequencing platforms, which target partial regions of the 16S rRNA gene, are widely used to reduce the cost burden of next-generation sequencing (NGS). However, 16S rRNA gene sequencing detects only part of the microbiota community revealed by shotgun sequencing, which has a greater ability to detect less abundant taxa. These less abundant taxa may hold significant biological relevance [135]. In the case of shotgun metagenomic sequences, it is important to note that non-microbial reads, such as host DNA or other contaminants, can become overrepresented and obscure the microbiome data. This issue is particularly noticeable in human reproductive tract microbiome studies, such as those using vaginal swabs or endometrial biopsy samples, which often contain significant amounts of human DNA [133].

The taxonomic names generated by community sequencing are not always reliable. Taxonomy is assigned by comparing sequencing reads to a reference database, but the algorithms used for this process are imperfect [136,137]. An even more significant issue is the inaccuracy and incompleteness of reference databases, particularly when general-purpose databases are used. These databases broadly capture microbial diversity but are not specifically optimized for the vaginal microbiome. While some vaginal-specific databases have been developed, there remains a need for further improvements in this area [63,138]. The relative abundances obtained through community sequencing often differ from the true relative abundances, as each step in the process can bias the detection of certain species over others. Such biases are protocol-specific, making quantitative comparisons between studies using different protocols unreliable [139].

Considering the advantages and disadvantages of 16S rRNA and metagenomic sequencing, novel technologies are emerging which could combine the benefits of both technologies in the future. This might lead to more accurate methods that can help define the microbiota pattern of the female genital tract and help to deepen our understanding of its association with various obstetric and gynecological conditions.

## 9. Limitations

The current scientific knowledge is contradictory and shares some limitations regarding the relationship between an altered vaginal or endometrial microbiome and various gynecological and pregnancy-related conditions. These discrepancies generally arise from the fact that different studies apply varying methodological aspects and sample or case–control group selection criteria. Furthermore, it is known that the bacterial community pattern can differ among ethnic groups and may also dynamically change during the menstrual cycle. When comparing individual studies, the sampling must be taken into account. It is critical to distinguish whether an endometrial biopsy or an endometrial fluid sample was taken; in addition, in the case of vaginal samples, self-sampling versus sampling by healthcare professionals is an additional factor to consider. Additionally, the comparability of studies is hindered by differences in sample processing, DNA extraction, library preparation, the use of positive and negative controls, sequencing, and bioinformatics analysis. Finally, it is crucial to emphasize that socioeconomic differences, individual genetic predispositions, and history of medical treatments can also influence the current composition of the microbiome and its categorization as normal or dysbiotic. Beyond adopting standardized methodologies, it will be important in the future to conduct more longitudinal studies that can monitor changes in the reproductive tract microbiome over time in specific groups.

Nowadays, comprehensive reviews are available in the field of genital microbiome technologies, providing a framework for developing standardized guidelines. These methodological recommendations outline the minimum standard requirements and guidance that researchers must follow in human microbiome studies to ensure best practices. Furthermore, they highlight critical pitfalls to avoid in order to conduct high-quality microbiome research in the area of the female genital microbiome and to develop standardized tests [128,140].

## 10. Conclusions and Recommendations

With regard to the application of the female genital microbiome as an accurate diagnostic biomarker, most studies currently confirm the protective effect of *Lactobacillus* spp. abundance in relation to gestational age and fertility. Additional results have indicated that persistent infections in the vagina or uterine cavity may increase the risk of early pregnancy loss and lead to unfavorable outcomes. Genital microbial imbalances or alterations in the microbiome pattern have been implicated in disrupting the mucosal barrier, facilitating the penetration of foreign immunogenic antigens and the inappropriate activation of pro-inflammatory networks. However, according to currently available data, there is no definitive microbial biomarker that can be clearly associated with preterm delivery, recurrent pregnancy loss, or embryo implantation failure, or that can diagnose or monitor infertile status.

Previous studies have demonstrated that molecular microbiology is a reliable, fast, and efficient diagnostic tool that enables the detection of both culturable and non-culturable bacteria associated with chronic endometritis. It demonstrates a 77% concordance with traditional diagnostic methods, such as histology, hysteroscopy, and microbial culture [141]. This is particularly important, as CE can be asymptomatic and may affect up to 40% of individuals experiencing infertility issues. It can be responsible for recurrent implantation failures and recurrent miscarriages. According to current expert opinions, routine CE screening of all patients with infertility is not recommended; however, the testing of RIF and RPL patients should be considered, especially in cases of proven euploid miscarriage. Studies have shown improved pregnancy and live birth rates in women with chronic endometritis following successful antibiotic treatment [97,142].

The relationship between the vaginal microbiome and fertility or pregnancy is an area of intense research. The protective role of a normal vaginal microbiome is well established in protecting against bacterial vaginosis, urinary tract infections, and sexually transmitted infections. Based on an analysis of the vaginal microbiome, susceptibility to sexually transmitted infections and pelvic inflammatory disease can be indirectly estimated. A European team of 37 fertility experts and clinicians has issued recommendations for the use of vaginal microbiome testing, particularly in the field of assisted reproduction [143]. At present, the evidence is too limited to recommend a routine screening of vaginal dysbiosis in all patients suffering infertility conditions. However, other discussions suggest routine vaginal microbiota testing for all women undergoing infertility treatment. One of the key conclusions from the recommendations is that there is no current gold standard for assessing vaginal dysbiosis.

Multiple techniques have been developed for evaluation and characterization, including the Amsel criteria, Nugent scoring, pH testing, and qPCR methods. Even amidst high microbial complexity, the characterization of DNA/RNA sequences allows for the identification of low-abundance or difficult-to-culture bacteria. NGS technology has enhanced the resolution of testing and accelerated research, allowing for the tracking of entire complex microbiome communities. Recent studies indicate that even small changes in the vaginal microbiome can significantly influence fertility, which calls for more specific investigations to develop precise therapeutic protocols for infertility treatments.

In terms of future directions, a clear definition of the normal microbiome in a healthy reproductive tract is essential. Although fundamental findings have emerged from research, the data are still insufficient to definitively characterize healthy versus dysbiotic states. The first step toward this goal involves standardizing protocols, sampling methods, sequencing techniques, and bioinformatics analyses. This will provide comparable data and may lead to general conclusions that serve as the foundation for future effective therapies. Additionally, the development of advanced diagnostic tools and the identification of specific biomarkers will be crucial to improving the accuracy of diagnoses and personalizing treatments. Such a novel approach to biomarker identification can be seen in the field of reproductive medicine, where the presence of specific miRNA biomarkers can correlate with certain microbiome signatures [24]. Proteomic, metabolomic, and transcriptomic analyses, combined with metagenomic investigations of well-characterized vaginal and endometrial microbiome samples, are essential to elucidate the pathogenic signatures linked to gynecological diseases driven by dysbiosis, and are also critical for identifying medically applicable biomarkers. This multi-omics approach can help to decipher additional drivers of health and disease in the genital microbiome. Continued research, including large-scale clinical studies, will be necessary to further refine therapeutic protocols and address challenges related to individual variability and the complexity of the microbiome.

## Figures and Tables

**Figure 1 ijms-25-13227-f001:**
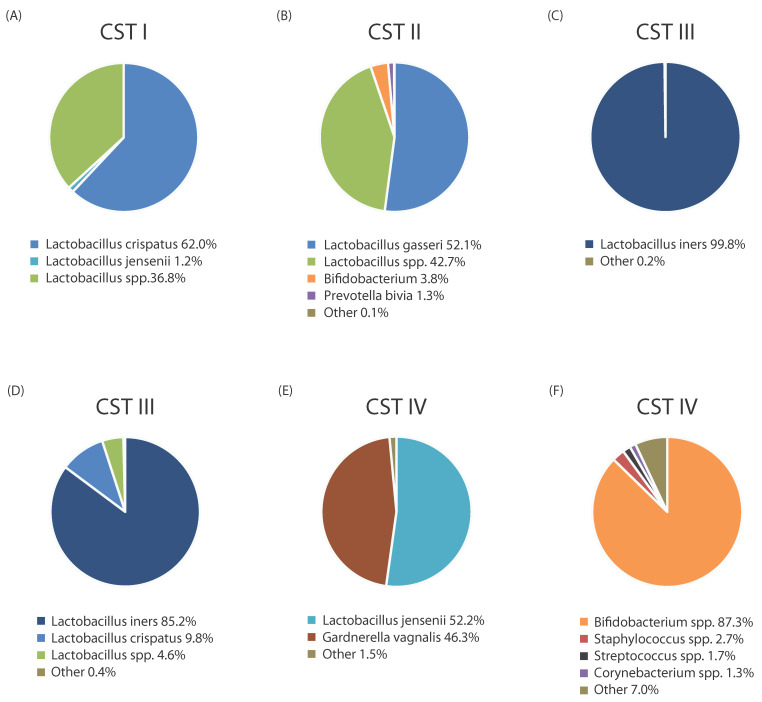
Our unpublished data represent different vaginal community state types (CSTs) of reproductive-aged asymptomatic women from vaginal swab samples. (**A**) is CST I (**B**) is CST II, (**C**) and (**D**) are CST III, (**E**) and (**F**) are CST IV. CSTs of the vaginal microbiome were classified according to the VALENCIA method by France MT et al. (2020) [18]. The results were obtained using the Illumina MiSeq platform, based on next-generation sequencing analysis of the bacterial 16S rRNA hypervariable regions (V1-V3 and V3-V5).

**Figure 2 ijms-25-13227-f002:**
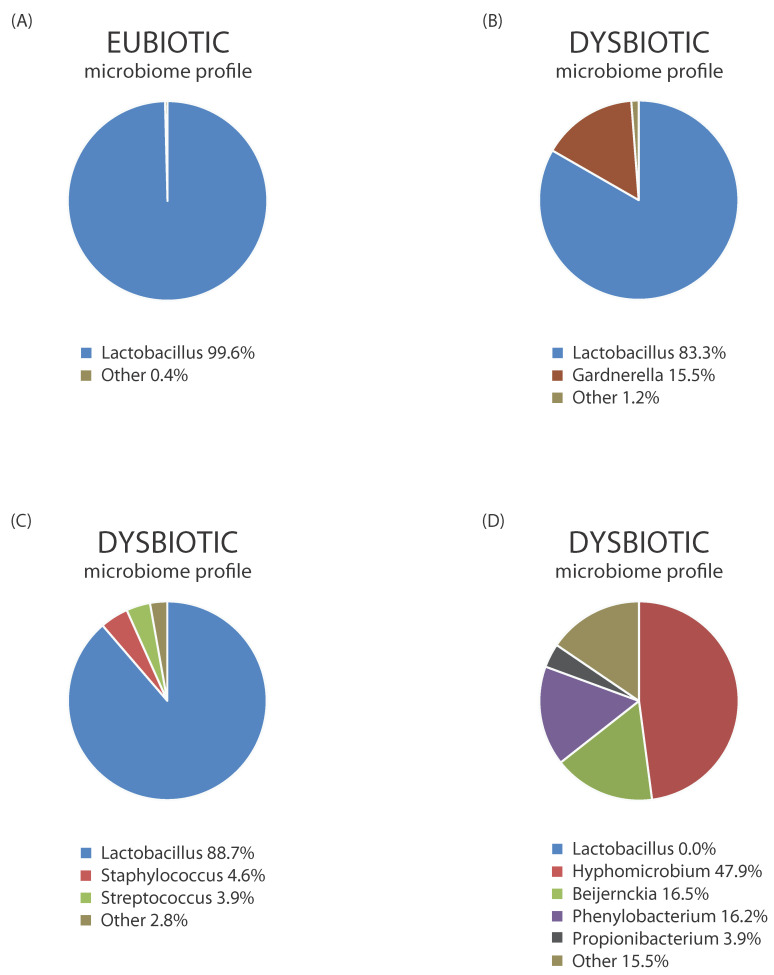
Bacterial profiles of the eubiotic (**A**) and various dysbiotic states (**B**,**C**,**D**) of the endometrial microbiome based on next-generation sequencing (NGS) analysis targeting bacterial 16S rRNA hypervariable regions (V2, V3, V4, V6, V7, V8, V9), performed on the Ion Torrent S5 platform. The figure presents our unpublished results obtained from endometrial biopsy samples.

**Table 1 ijms-25-13227-t001:** Summary of the main aspects of 16S rRNA sequencing and shotgun metagenomic sequencing.

Aspects/Features	16S rRNA Sequencing	Shotgun Metagenomic Sequencing
Cost	Cheaper	More expensive (depends on required sequencing depth)
Taxonomic resolution	Genus or species (depends on targeted hypervariable regions)	Species (depends on sequencing coverage)
Taxonomic coverage	Bacteria and archaea	All taxa
Bias	Medium to high (identified taxonomic composition depends on selected hypervariable regions)	Lower
Bioinformatics requirements	Beginner to intermediate expertise	Intermediate to advanced expertise
Databases	Established, well curated	Relatively new, still growing
Sensitivity to host DNA contamination	Low (depends on the presence of inhibitors and detectable microbiome)	High, varies with sample type (can be addressed by adjusting the sequencing depth)
Profile microbial genes—functional profiling	No	Yes

## Data Availability

No new data were created or analyzed in this study. Data sharing is not applicable to this article.

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
