# Peer review of "The Role of the Vaginal and Endometrial Microbiomes in Infertility and Their Impact on Pregnancy Outcomes in Light of Recent Literature"

_ijms, 2024, doi:10.3390/ijms252313227_

Round 1
Reviewer 1 Report
Comments and Suggestions for Authors
The article presents highly relevant information. However, it sometimes appears poorly structured and would benefit from further development on certain topics. The information is not always cohesively aligned across topics, and authors should pay close attention to the main objective of the study, ensuring that the literature review remains focused on the central theme of the article. At times, parts of the manuscript appear overly general; refining the content to maintain a clear focus would increase the overall impact and clarity of the work.
Although the information in the abstract is clear and consistent, its content does not reflect the central theme of the article. Authors should value the content of the abstract and directly emphasize the relationship between microbiomes, infertility and pregnancy, as these are the main focus of the article. It is necessary to explicitly mention infertility and its impacts.
In the topic “The vaginal microbiome and its impact on fertility” the authors should clarify the role of the endometrial microbiome in fertility, as it receives little emphasis. Although the vaginal microbiome is correctly and extensively explored, as the title suggests, the work focuses on the relationship between the vaginal and endometrial microbiomes in fertility and pregancy.
Throughout the text, the link between vaginal dysbiosis and infertility is unclear. It would be important to expand the influence of different dysbiotic states on fertility and the success of fertility treatments. Authors should clarify this point in the manuscript.
Despite the extensive description of CSTs, the authors do not relate it to the main focus of the article. For example, rather than discussing the proportions of each microorganism in each CST, it would be more important to relate and link these CSTs (particularly CST III and CST IV) to positive and negative outcomes in the domain of fertility and pregnancy. Authors should expand this discussion in the article to increase its impact.
The authors clearly mention the various risks during pregnancy such as premature birth, infection, and pregnancy loss. However, it would be valuable to integrate a discussion on how the microbiome contributes to these outcomes by focusing on inflammatory processes and immune responses.
In the final part of the section, the authors highlight the relevance of the microbiome in idiopathic infertility, which is indeed a highly relevant topic. However, authors should emphasize the potential of the microbiome as a biomarker or even as a therapeutic target to strengthen the link between the microbiome and potential treatment advances.
Regarding the topic “Vaginal microbiome and pregnancy”, the manuscript links microbial diversity with complications in pregnancy, including premature birth and miscarriage. However, the authors must highlight the importance of these findings for diagnosis and monitoring in clinical practice.
The authors raise a highly relevant discussion regarding the contradictory results in the literature on TSC (specifically TSC III-V) and its association with preterm birth. This point is indeed very important, as it highlights some scientific and experimental uncertainties in certain studies. Authors should add additional discussion of possible reasons for these divergent conclusions (whether methodological, population selection, or other factors).
Regarding the species L. iners, the manuscript states that the presence of this species may play an unfavorable role in implantation and possible unexplained infertility. It is crucial that the authors expand the discussion focused on the role of L. iners. Furthermore, in contrast, the species L. crispatus could be discussed from the point of view of benefits for vaginal health. Authors must pay attention to the content presented and its connection with the first topic.
The authors suggest antimicrobial testing before conception and early in pregnancy to predict positive reproductive outcomes. However, the limitations and challenges of this practice are not clearly addressed. Authors should explore and add this information to the manuscript.
In relation to the topic “The endometrial microbiome and its role in fertility”, the manuscript explores the endometrial microbiome and how it can directly affect fertility and reproduction. Dysbiosis, characterized by the presence of Gardnerella, Atopobium and Prevotella, is associated with reproductive difficulties. The authors should expand on this topic and discuss how dysbiosis directly affects the endometrial mucosal barrier and immune responses, focusing on the mechanisms involved. Similar to the previous topic, the authors cite some contradictory conclusions presented in studies on the microbial composition of the endometrium and its impact on fertility. As suggested in the previous topic, authors should discuss possible methodological limitations or variability between studies.
The authors indicate the presence of Lactobacillus as a biomarker for IVF success. However, the authors must discuss the challenges of clinical implementation of this type of biomarker and its standardization in terms of analysis.
The topic “Endometrial Microbiome and Pregnancy” explores a potential relationship between a Lactobacillus-dominant microbiome and reproductive success, as it may promote a favorable environment for implantation and immune modulation. On the other hand, an NLD microbiome can trigger an inflammatory response that can interfere with embryo implantation. The authors could explore how pathogenic bacteria, such as Streptococcus and Klebsiella, can affect the endometrial barrier and promote inflammation. Similar to previous topics, the authors discuss the potential of using microbiome studies to identify pathogenic bacteria and thus improve diagnosis. This is indeed a promising point that the authors should develop further, especially in the context of assisted fertility treatments.
In the topic “Interactions between the endometrial and vaginal microbiomes” the authors report that the vagina has lower microbial diversity compared to the endometrium. Authors should discuss these differences considering ecological and anatomical factors. The authors indicate that vaginal dysbiosis may contribute to endometrial dysbiosis due to pro-inflammatory responses and damage to the cervical epithelial barrier. However, as this is in fact the central focus of the manuscript, the authors should expand the discussion on the mechanisms and inflammatory responses that justify bacterial translocation to the endometrium.
The authors discuss the importance of both microbiomes in relation to infertility. However, overall, the authors could strengthen this topic further as it is the central focus of the manuscript.
The numbering of topics in the manuscript goes from point 5 to point 7.
The topic “Basic treatment options for endometrial and vaginal dysbiosis” addresses different therapeutic approaches for treating dysbiosis. In addition to the use of lactic acid and probiotics, the authors discuss the effectiveness of antibiotics in treating CE. However, the authors must highlight the difficulty in diagnosing CE, especially because the majority of cases are asymptomatic. Furthermore, the authors discuss microbiota transplantation as a way to restore eubiosis in patients who do not respond to conventional treatments. However, authors should include a discussion of the lack of standardized protocols, donor selection criteria, as well as potential ethical challenges.
The numbering of the manuscript topics goes from point 8 to point 10.
The topic "Modern examination methods for the vaginal and endometrial microbiomes" is appropriately framed within the overall context of the article, as it comprehensively addresses advanced methodologies used to analyze the vaginal and endometrial microbiomes, which are crucial for understanding their influences on fertility and reproduction health. The inclusion of a comparison between 16S rRNA and metagenomic sequencing is relevant, as it highlights the advantages and limitations of each technique, elucidating how these approaches can contribute to the identification of states of eubiosis and dysbiosis. I do not suggest any changes to this topic.
As far as conclusions are concerned, they could be further strengthened by placing additional emphasis on the importance of integrating clinical trial data. Additionally, the mention of “challenges related to individual variability and microbiome complexity” could be expanded to include specific proposals to address them, which would increase the overall impact of the manuscript.
Author Response
Thank you for reviewing our manuscript entitled „The Role of the Vaginal and Endometrial Microbiomes in Infertility and Their Impact on Pregnancy Outcomes in the light of recent literature” (Manuscript number: IJMS-3297539). We have extensively revised the manuscript based on the criticisms of the Reviewer. Our answers are the following:
Reviewer 1:
The article presents highly relevant information. However, it sometimes appears poorly structured and would benefit from further development on certain topics. The information is not always cohesively aligned across topics, and authors should pay close attention to the main objective of the study, ensuring that the literature review remains focused on the central theme of the article. At times, parts of the manuscript appear overly general; refining the content to maintain a clear focus would increase the overall impact and clarity of the work.
Comment 1: Although the information in the abstract is clear and consistent, its content does not reflect the central theme of the article. Authors should value the content of the abstract and directly emphasize the relationship between microbiomes, infertility and pregnancy, as these are the main focus of the article. It is necessary to explicitly mention infertility and its impacts.
Response 1: Thank you for your comment. On this basis, we have added an introduction section to our manuscript, which presents the definition of infertility, its worldwide importance, prevalence, and main causes, emphasizing the fact of its association with the female genital microbiome, the detailed description of which is the aim of our study. In addition to this, several changes have been made in the manuscript to further emphasize the link between microbiomes, infertility and pregnancy.
Comment 2: In the topic “The vaginal microbiome and its impact on fertility” the authors should clarify the role of the endometrial microbiome in fertility, as it receives little emphasis. Although the vaginal microbiome is correctly and extensively explored, as the title suggests, the work focuses on the relationship between the vaginal and endometrial microbiomes in fertility and pregancy.
Throughout the text, the link between vaginal dysbiosis and infertility is unclear. It would be important to expand the influence of different dysbiotic states on fertility and the success of fertility treatments. Authors should clarify this point in the manuscript.
Response 2: In response to this suggestion, this chapter has been supplemented with additional findings on the association between vaginal dysbiosis and infertility.
Comment 3: Despite the extensive description of CSTs, the authors do not relate it to the main focus of the article. For example, rather than discussing the proportions of each microorganism in each CST, it would be more important to relate and link these CSTs (particularly CST III and CST IV) to positive and negative outcomes in the domain of fertility and pregnancy. Authors should expand this discussion in the article to increase its impact.
Response 3: We agree with this view and have devoted an additional paragraph in chapter 2.1 to discussing and evaluating the positive and negative effects of CSTs on fertility.
Comment 4: The authors clearly mention the various risks during pregnancy such as premature birth, infection, and pregnancy loss. However, it would be valuable to integrate a discussion on how the microbiome contributes to these outcomes by focusing on inflammatory processes and immune responses.
Response 4: Thank you for your remark, we have modified the manuscript to introduce and highlight the requested changes focusing on immunological processes.
Comment 5: In the final part of the section, the authors highlight the relevance of the microbiome in idiopathic infertility, which is indeed a highly relevant topic. However, authors should emphasize the potential of the microbiome as a biomarker or even as a therapeutic target to strengthen the link between the microbiome and potential treatment advances.
Response 5: Although the thousands of individual species that form the microbiome - with the exception of pathogenic species and some Lactobascillus species, which are discussed separately - are not yet very useful as accurate biomarkers. That said, combinations and patterns of these species (such as CSTs, among others) appear to be potential biomarkers for certain pathologies, but further refinement and diversification is essential. We have added this thought to the conclusion section of our literature review.
Comment 6: Regarding the topic “Vaginal microbiome and pregnancy”, the manuscript links microbial diversity with complications in pregnancy, including premature birth and miscarriage. However, the authors must highlight the importance of these findings for diagnosis and monitoring in clinical practice.
Response 6: The conclusion section was complemented by the discussion of the following topics as you suggested; i) the importance of microbiome findings for diagnosis and monitoring in clinical practice; ii) challenges for the clinical introduction of microbiome biomarkers and standardization of analysis; iii) the potential of microbiome tests to identify pathogenic bacteria and thus improve diagnosis, especially in the context of assisted fertility treatments.
Comment 7: The authors raise a highly relevant discussion regarding the contradictory results in the literature on TSC (specifically TSC III-V) and its association with preterm birth. This point is indeed very important, as it highlights some scientific and experimental uncertainties in certain studies. Authors should add additional discussion of possible reasons for these divergent conclusions (whether methodological, population selection, or other factors).
Response 7: Thank you for your valuable comments, the topics of these methodological, study population selections or sampling differences and their limitations in the research field have been added to the conclusion section of the manuscript.
Comment 8: Regarding the species L. iners, the manuscript states that the presence of this species may play an unfavorable role in implantation and possible unexplained infertility. It is crucial that the authors expand the discussion focused on the role of L. iners. Furthermore, in contrast, the species L. crispatus could be discussed from the point of view of benefits for vaginal health. Authors must pay attention to the content presented and its connection with the first topic.
Response 8: Following your important recommendation, in the chapter of “The vaginal microbiome and its impact on fertility”, we have described a more extensive and detailed characterization of L. iners and L. crispatus species.
Comment 9: The authors suggest antimicrobial testing before conception and early in pregnancy to predict positive reproductive outcomes. However, the limitations and challenges of this practice are not clearly addressed. Authors should explore and add this information to the manuscript.
Response 9: More information on the limitations of microbiome tests and their use in clinical practice is provided in the conclusion section.
Comment 10: In relation to the topic “The endometrial microbiome and its role in fertility”, the manuscript explores the endometrial microbiome and how it can directly affect fertility and reproduction. Dysbiosis, characterized by the presence of Gardnerella, Atopobium and Prevotella, is associated with reproductive difficulties. The authors should expand on this topic and discuss how dysbiosis directly affects the endometrial mucosal barrier and immune responses, focusing on the mechanisms involved.
Response 10: On request, we have elaborated on this topic in more detail and added further literatures to the chapter.
Comment 11: Similar to the previous topic, the authors cite some contradictory conclusions presented in studies on the microbial composition of the endometrium and its impact on fertility. As suggested in the previous topic, authors should discuss possible methodological limitations or variability between studies.
Response 11: Thank you for your valuable comments, the topics of these methodological, study population selections or sampling differences and their limitations in the research field have been added to the conclusion section of the manuscript.
Comment 12: The authors indicate the presence of Lactobacillus as a biomarker for IVF success. However, the authors must discuss the challenges of clinical implementation of this type of biomarker and its standardization in terms of analysis.
Response 12: The conclusion section was complemented by the discussion of the following topics as you suggested; i) the importance of microbiome findings for diagnosis and monitoring in clinical practice; ii) challenges for the clinical introduction of microbiome biomarkers and standardization of analysis; iii) the potential of microbiome tests to identify pathogenic bacteria and thus improve diagnosis, especially in the context of assisted fertility treatments.
Comment 13: The topic “Endometrial Microbiome and Pregnancy” explores a potential relationship between a Lactobacillus-dominant microbiome and reproductive success, as it may promote a favorable environment for implantation and immune modulation. On the other hand, an NLD microbiome can trigger an inflammatory response that can interfere with embryo implantation. The authors could explore how pathogenic bacteria, such as Streptococcus and Klebsiella, can affect the endometrial barrier and promote inflammation. Similar to previous topics, the authors discuss the potential of using microbiome studies to identify pathogenic bacteria and thus improve diagnosis. This is indeed a promising point that the authors should develop further, especially in the context of assisted fertility treatments.
Response 13: The conclusion section was complemented by the discussion of the following topics as you suggested; i) the importance of microbiome findings for diagnosis and monitoring in clinical practice; ii) challenges for the clinical introduction of microbiome biomarkers and standardization of analysis; iii) the potential of microbiome tests to identify pathogenic bacteria and thus improve diagnosis, especially in the context of assisted fertility treatments.
Comment 14: In the topic “Interactions between the endometrial and vaginal microbiomes” the authors report that the vagina has lower microbial diversity compared to the endometrium. Authors should discuss these differences considering ecological and anatomical factors. The authors indicate that vaginal dysbiosis may contribute to endometrial dysbiosis due to pro-inflammatory responses and damage to the cervical epithelial barrier. However, as this is in fact the central focus of the manuscript, the authors should expand the discussion on the mechanisms and inflammatory responses that justify bacterial translocation to the endometrium.
Response 14: Thank you for your comments, in chapters 2 and 4 on the vaginal and endometrial microbiome we also wrote about the relationship between the microbiome and the immune system or the inflammation-triggering effects of dysbiosis, but we have added further information about this process in this section.
The authors discuss the importance of both microbiomes in relation to infertility. However, overall, the authors could strengthen this topic further as it is the central focus of the manuscript.
Comment 15: The numbering of topics in the manuscript goes from point 5 to point 7.
Response 15: Thank you for your remark, we have corrected the chapter numbering.
Comment 16: The topic “Basic treatment options for endometrial and vaginal dysbiosis” addresses different therapeutic approaches for treating dysbiosis. In addition to the use of lactic acid and probiotics, the authors discuss the effectiveness of antibiotics in treating CE. However, the authors must highlight the difficulty in diagnosing CE, especially because the majority of cases are asymptomatic.
Response 16: We agree with the proposition and we have mentioned the diagnostic challenges of CE in this chapter.
Comment 17: Furthermore, the authors discuss microbiota transplantation as a way to restore eubiosis in patients who do not respond to conventional treatments. However, authors should include a discussion of the lack of standardized protocols, donor selection criteria, as well as potential ethical challenges.
Response 17: We are dealing with a very important and future-oriented issue, so we presented the microbiota transplantation in a more comprehensive way, explaining its limitations.
Comment 18: The numbering of the manuscript topics goes from point 8 to point 10.
Response 18: Thank you for your remark, we have corrected the chapter numbering.
Comment 19: The topic "Modern examination methods for the vaginal and endometrial microbiomes" is appropriately framed within the overall context of the article, as it comprehensively addresses advanced methodologies used to analyze the vaginal and endometrial microbiomes, which are crucial for understanding their influences on fertility and reproduction health. The inclusion of a comparison between 16S rRNA and metagenomic sequencing is relevant, as it highlights the advantages and limitations of each technique, elucidating how these approaches can contribute to the identification of states of eubiosis and dysbiosis. I do not suggest any changes to this topic.
Comment 20: As far as conclusions are concerned, they could be further strengthened by placing additional emphasis on the importance of integrating clinical trial data. Additionally, the mention of “challenges related to individual variability and microbiome complexity” could be expanded to include specific proposals to address them, which would increase the overall impact of the manuscript.
Response 20: On the basis of these suggestions, we have inserted several additional topics into the conclusion of our review article, which we discuss in more detail. With these changes, we hope to have made our review even more valuable and a more relevant in the light of the current literature.
Finally, we would like to express our gratitude to the Reviewer for the remarks that have made our paper much more valuable. We hope that the extensive changes we made in the manuscript will be to your satisfaction, and now you will find our work to be worthy of publication.
Reviewer 2 Report
Comments and Suggestions for Authors
Review
Thanks for the paper that is submitted, which covers an interesting topic. I think this article attempts to present a comprehensive analysis of the relationship between the vaginal and endometrial microbiome with fertility and pregnancy outcomes.
I have major comments and edits that I would like to suggest to the authors. After using an anti-plagiarism tool, I have found, literally copied, some paragraphs from this article up to 3%:
Toson B, Simon C, Moreno I. The Endometrial Microbiome and Its Impact on Human Conception. Int J Mol Sci. 2022;23(1):485. Published 2022 January 1. doi:10.3390/ijms23010485
I encourage the authors to modify the manuscript to reduce this percentage.
Title: The title is misleading because we do not know what the authors have done. I suggest introducing in the title the type of manuscript: Literature review for example...
If it is a Scoping review, the authors should include the PRISMA ScR check list, and also the flow diagram, search strategy….
There are no sections that structure the article such us: introduction, methods...
The authors should follow the recommendations of the journal with regard to headings and subheadings such as section 1 and 3.
The thematic organization is clear, with sections on the impact on fertility, pregnancy and reproductive complications. However, it might be useful to reorganize some sections to address first the anatomy and function of the vaginal and endometrial microbiome, followed by its relationship to infertility and pregnancy complications.
There are some long and complex sentences that could be broken up to improve readability.
There is some unnecessary repetition of information, especially at the beginning of the manuscript.
It is recommended that terms such as “vaginal microbiome” and “vaginal flora” be standardized to maintain consistency throughout the document.
Including graphs and tables that summarize key data can improve understanding and visualization of the information.
Some inconsistencies are observed in the use of capital letters, especially in the names of bacterial species. Please homogenize them.
Being more specific:
Lines 128-130/145-146: The font size is slightly different from the rest of the manuscript. Correct the font size differences in all the document.
Heading number 4:
Although the differentiation between the vaginal and endometrial microbiomes is mentioned, a more in-depth discussion of how anatomical and functional differences between the two environments may influence the colonization and stability of these microbiomes in terms of reproductive health is lacking. Furthermore, the section on the endometrial microbiome is less detailed compared to the section on the vaginal microbiome.
Heading 6 and 9 are missing!!
Heading number 10:
Although the limitations of current technologies are mentioned, the discussion of specific biases (such as selection bias in microbiome studies) and how these may affect data interpretation could be expanded. Further mention of current gaps in the literature and recommendations for future research could improve the conclusion. This would be particularly relevant in the area of microbiota-based treatments, where evidence is still limited.
References:
Citations are well distributed throughout the article, but it would be beneficial to include recent studies, especially in heading eight related to emerging treatments and microbiome sequencing technologies. Some references are outdated (such as 103-104-105….)
Author Response
Thank you for reviewing our manuscript entitled „The Role of the Vaginal and Endometrial Microbiomes in Infertility and Their Impact on Pregnancy Outcomes in the light of recent literature” (Manuscript number: IJMS-3297539). We have extensively revised the manuscript based on the criticisms of the Reviewer. Our answers are the following:
Review 2:
Thanks for the paper that is submitted, which covers an interesting topic. I think this article attempts to present a comprehensive analysis of the relationship between the vaginal and endometrial microbiome with fertility and pregnancy outcomes.
Comment 1: I have major comments and edits that I would like to suggest to the authors.
After using an anti-plagiarism tool, I have found, literally copied, some paragraphs from this article up to 3%:
Toson B, Simon C, Moreno I. The Endometrial Microbiome and Its Impact on Human Conception. Int J Mol Sci. 2022;23(1):485. Published 2022 January 1. doi:10.3390/ijms23010485
I encourage the authors to modify the manuscript to reduce this percentage.
Response 1: Thank you for your comment. Throughout our work, including the preparation of this manuscript, we have taken great care to avoid plagiarism. We acknowledge that in a literary review article, there are instances where retaining the original wording may be necessary. Utilizing our proprietary AI-based document comparison system, we found no similarities between our complete manuscript and the referenced articles. However, it is possible that something may have escaped our attention. Therefore, we kindly request that you provide us with the specific instances of overlap so that we can address and rectify them accordingly.
Comment 2: Title: The title is misleading because we do not know what the authors have done. I suggest introducing in the title the type of manuscript: Literature review for example...
Response 2: Thank you for your suggestion, we have modified the title of our manuscript accordingly.
Comment 3: If it is a Scoping review, the authors should include the PRISMA ScR check list, and also the flow diagram, search strategy….
Response 3: Our work is not a scoping review, but a literature review, so we do not use a PRISMA ScR checklist.
Comment 4: There are no sections that structure the article such us: introduction, methods...
The authors should follow the recommendations of the journal with regard to headings and subheadings such as section 1 and 3.
Response 3: Thanks for your insight, we have further sectioned the chapters as you recommended and added an introduction section.
Comment 5: The thematic organization is clear, with sections on the impact on fertility, pregnancy and reproductive complications. However, it might be useful to reorganize some sections to address first the anatomy and function of the vaginal and endometrial microbiome, followed by its relationship to infertility and pregnancy complications.
Response 5: Thank you for your suggestion. We have divided the chapters describing both the vaginal and the endometrial microbiome into a general overview section, which includes a description of the bacterial community types, the dynamic physiological changes of the microbiome, and the factors that influence its functioning. This is followed by the second part, where the literature background on the relationship between the microbiome and fertility and IVF treatment outcomes is presented.
Comment 6: There are some long and complex sentences that could be broken up to improve readability.
There is some unnecessary repetition of information, especially at the beginning of the manuscript.
Response 6: We have tried to simplify the sentences as requested.
Comment 7: It is recommended that terms such as “vaginal microbiome” and “vaginal flora” be standardized to maintain consistency throughout the document.
Response 7: We have corrected the manuscript accordingly.
Comment 8: Including graphs and tables that summarize key data can improve understanding and visualization of the information.
Response 8: Thank you for your feedback. In our literary review manuscript, it was essential to focus on selecting from the numerous potential graphics and charts. We agree that the data could be represented in many other visual formats; however, considering the current constraints, we trust that the chosen representations will still be helpful to readers.
Comment 9: Some inconsistencies are observed in the use of capital letters, especially in the names of bacterial species. Please homogenize them.
Response 9: We have modified the manuscript accordingly.
Being more specific:
Comment 10: Lines 128-130/145-146: The font size is slightly different from the rest of the manuscript. Correct the font size differences in all the document.
Response 10: We have modified the manuscript accordingly.
Comment 11: Heading number 4:
Although the differentiation between the vaginal and endometrial microbiomes is mentioned, a more in-depth discussion of how anatomical and functional differences between the two environments may influence the colonization and stability of these microbiomes in terms of reproductive health is lacking. Furthermore, the section on the endometrial microbiome is less detailed compared to the section on the vaginal microbiome.
Response 11: Thank you for your important comment, after the revision several additional related topics were presented in the chapter “The vaginal microbiome and its impact on fertility” e.g. more detailed characterization of Lactobacillus types, general immune processes and inflammatory responses in relation to dysbiosis, physiological changes of the microbiome, therefore we have only made a shorter addition in this section.
Comment 12: Heading 6 and 9 are missing!!
Response 12: We have corrected the numbering of the chapters.
Comment 13: Heading number 10:
Although the limitations of current technologies are mentioned, the discussion of specific biases (such as selection bias in microbiome studies) and how these may affect data interpretation could be expanded. Further mention of current gaps in the literature and recommendations for future research could improve the conclusion. This would be particularly relevant in the area of microbiota-based treatments, where evidence is still limited.
Response 13: Thanks for your valuable comment, we have corrected the conclusion of our manuscript accordingly. The topics of these methodological, study population selections or sampling differences and their limitations in the research field have been added to the conclusion section. In addition we discuss about the challenges for the clinical introduction of microbiome biomarkers and standardization of analysis.
In summary, on the basis of these suggestions, we have inserted several additional topics into the conclusion of our review article, which we discuss in more detail. With these changes, we hope to have made our review even more valuable and a more relevant in the light of the current literature.
Comment 14: References:
Citations are well distributed throughout the article, but it would be beneficial to include recent studies, especially in heading eight related to emerging treatments and microbiome sequencing technologies. Some references are outdated (such as 103-104-105….)
Response 14: Additional recent literatures have been cited for the relevant part of our review article.
Finally, we would like to express our gratitude to the Reviewer for the remarks that have made our paper much more valuable. We hope that the extensive changes we made in the manuscript will be to your satisfaction, and now you will find our work to be worthy of publication.
Round 2
Reviewer 2 Report
Comments and Suggestions for Authors
Dear authors,
- Thank you for taking the time to address comments on the manuscript. The article provides a thorough overview of the role of vaginal and endometrial microbiomes in infertility and their impact on pregnancy outcomes. It effectively synthesizes recent research and includes diverse studies. After the revision the paper is well-organized, with sections covering various microbiome components and their implications for fertility and pregnancy. Although the manuscript has been greatly improved, but there are still some concerns about some issues:
-Regarding the plagiarism issue, I have asked the editors to submit the plagiarism report to the authors.
-The abstract is dense and could better summarize the key findings and clinical implications of the review.
-Some sections, especially on the diagnostic technologies and microbial diversity, repeat similar information, which can be streamlined.
-Recommendations for future research and clinical protocols are scattered; a dedicated conclusion or "Recommendations" section would improve coherence.
-I suggest the authors to include the studies limitations or contradictory findings of the papers that conform the review. This issue is essential for a review.
Author Response
Thank you for taking the time to address comments on the manuscript. The article provides a thorough overview of the role of vaginal and endometrial microbiomes in infertility and their impact on pregnancy outcomes. It effectively synthesizes recent research and includes diverse studies. After the revision the paper is well-organized, with sections covering various microbiome components and their implications for fertility and pregnancy. Although the manuscript has been greatly improved, but there are still some concerns about some issues:
Comment 1: Regarding the plagiarism issue, I have asked the editors to submit the plagiarism report to the authors.
Response 1: Thank you for your feedback. We have received the plagiarism report from the editorial team, which indicated a very minimal amount of duplication. There were two sentences (approximately from lines 290 and 604 in the manuscript) that showed similarity to the cited reference. These sentences have been rephrased while retaining their original meaning. All other similarities were either irrelevant (e.g., "These authors have...") or necessary due to the technical terminology required for the content (e.g., "during menstrual cycle").
Comment 2: The abstract is dense and could better summarize the key findings and clinical implications of the review.
Response 2: Thank you for your suggestion, we have better specified the content of the abstract in relation to the clinical impact and applications of the endometrial and vaginal microbiome as well as the current state of research on this field.
Comment 3: Some sections, especially on the diagnostic technologies and microbial diversity, repeat similar information, which can be streamlined.
Response 3: Thank you for your comment, we have tried to further consolidate the relevant parts.
Comment 4: Recommendations for future research and clinical protocols are scattered; a dedicated conclusion or "Recommendations" section would improve coherence.
Response 4: The conclusion has been modified to include current recommendations for microbiome screening and testing, and to highlight opportunities for future research in this area.
Comment 5: I suggest the authors to include the studies limitations or contradictory findings of the papers that conform the review. This issue is essential for a review.
Response 5: Thank you for your remark. We have added a new ’limitations’ section to the review, in which we also present methodological recommendations for genital microbiome testing.
Once again, we are grateful for your constructive criticism and suggestions to improve the manuscript and hope it will be found suitable for publication.